# The Strong Anti-Tumor Effect of Smp24 in Lung Adenocarcinoma A549 Cells Depends on Its Induction of Mitochondrial Dysfunctions and ROS Accumulation

**DOI:** 10.3390/toxins14090590

**Published:** 2022-08-27

**Authors:** Ruiyin Guo, Xuewen Chen, Tienthanh Nguyen, Jinwei Chai, Yahua Gao, Jiena Wu, Jinqiao Li, Mohamed A. Abdel-Rahman, Xin Chen, Xueqing Xu

**Affiliations:** 1Department of Pulmonary and Critical Care Medicine, Zhujiang Hospital, Southern Medical University, Guangzhou 510280, China; 2Guangdong Provincial Key Laboratory of New Drug Screening, School of Pharmaceutical Sciences, Southern Medical University, Guangzhou 510515, China; 3Zoology Department, Faculty of Science, Suez Canal University, Ismailia 41522, Egypt

**Keywords:** scorpion venom, antimicrobial peptides, *Scorpio maurus palmatus*, Smp24, A549, apoptosis, autophagy, necrosis, cell cycle arrest

## Abstract

Non-small cell lung cancer (NSCLC) is the leading cause of death in lung cancer due to its aggressiveness and rapid migration. The potent antitumor effect of Smp24, an antimicrobial peptide derived from Egyptian scorpion *Scorpio maurus palmatus* via damaging the membrane and cytoskeleton have been reported earlier. However, its effects on mitochondrial functions and ROS accumulation in human lung cancer cells remain unknown. In the current study, we discovered that Smp24 can interact with the cell membrane and be internalized into A549 cells via endocytosis, followed by targeting mitochondria and affect mitochondrial function, which significantly causes ROS overproduction, altering mitochondrial membrane potential and the expression of cell cycle distribution-related proteins, mitochondrial apoptotic pathway, MAPK, as well as PI3K/Akt/mTOR/FAK signaling pathways. In summary, the antitumor effect of Smp24 against A549 cells is related to the induction of apoptosis, autophagy plus cell cycle arrest via mitochondrial dysfunction, and ROS accumulation. Accordingly, our findings shed light on the anticancer mechanism of Smp24, which may contribute to its further development as a potential agent in the treatment of lung cancer cells.

## 1. Introduction

Despite the effectiveness of traditional therapy in cancer treatment, such as chemotherapy, radiotherapy, and immune therapy, lung adenocarcinoma remains the major cause of cancer death due to its aggressiveness and rapid migration [1]. Besides, the cytotoxicity and drug resistance caused by those traditional therapy lead to the urgent need of alternative treatment with low cytotoxicity and treatment resistance.

In addition to the importance in the innate immune defense, antimicrobial peptides (AMPs) have emerged as promising drug candidates against multiple diseases, especially in anticancer therapy. Most of AMPs suppress the tumor cells by their direct interaction with the cell membrane rather than specific receptors, which prevents the drug resistance caused by other traditional treatment methods [2,3]. Notably, in addition to the membrane disruption of cancer cells, AMPs have been proven to exert an antitumor effect via targeting different cellular structures, interfering intrinsic pathways, which consequently results in abnormal changes in multiple events, such as apoptosis, autophagy, and cell cycle distribution [2,4]. Mitochondria are double-membrane-bound cell organelles found in most eukaryotic organisms that generate most of the chemical energy adenosine triphosphate (ATP). In addition to the production of ATP, the mitochondria have a major role in calcium ion storage and cellular proliferation regulation, such as apoptosis and cell cycle distribution regulation, making it a promising target in cancer treatments. There are abundant AMPs that have been reported to target mitochondria, leading to its dysfunction, which consequently alters the mitochondrial membrane potential and ROS production, inducing cancer cell apoptosis and cell cycle arrest. For instance, melittin, an amphiphilic alpha-helical peptide derived from honeybee venom (*Apis mellifera*), suppresses the proliferation of human gastric cancer cells via activating the mitochondrial pathway, thereby inducing the apoptosis process [5]. Previous study revealed the dedication of both necrosis and apoptosis in the antitumor mechanism of Brevinin-1RL1 by inducing extrinsic and mitochondria intrinsic apoptosis [6]. Moreover, the apoptosis of osteosarcoma MG63 cells is induced by MSP-4 via activation of Fas/FasL—and mitochondria-mediated pathway [7]. Interestingly, both pardaxin, an AMP isolated from *Pardachirus marmoratus*, and our previously reported peptide Smp43 share a similar mode of action in antitumor via inducing apoptosis and cell cycle arrest after disruption of mitochondrial membrane, both Smp43 and pardaxin also induce autophagy of hepatoma cells and ovarian cancer cells, respectively [8,9]. In previous study, we demonstrated the potent antitumor effect of Smp24 (IWSFLIKAATKLLPSLFGGGKKDS), another venom-derived AMP of *Scorpio maurus palmatus,* toward human non-small-cell lung cancer cell (NSCLC) A549. Notably, Smp24 caused mitochondrial damage which was represented by the release of calcein AM in flow cytometry analysis [10]. However, whether Smp24 has the above effects in human non-small-cell lung cancer cell remain elusive. In the current study, we investigated the interaction between Smp24 and mitochondria as well as its effect on apoptosis, cell cycle distribution, and autophagy. Our findings revealed that treatment with Smp24 led to the disruption of the mitochondrial membrane and accumulation of ROS, inducing apoptosis, cell cycle arrest, and autophagy of A549 cells. This discovery exposes the antitumor mechanism of Smp24 against NSCLC A549, suggesting its application in lung carcinoma therapy.

## 2. Results

### 2.1. Smp24 Suppresses the Proliferation of Human Lung Cancer Cells

As shown in Figure 1A, consistent with the previous report [10], Smp24 significantly suppressed the proliferation of A549 with the IC_50_ value at approximately 4.06 µM, and it exhibited less inhibitory potency toward normal cells, MRC-5, as evidenced from higher IC_50_ of approximately 14.68 ± 0.79 μM. Furthermore, a concentration- and time-dependent cytotoxicity toward A549 cells was induced by Smp24. Furtherly, the survival rate of Smp24-treated A549 cells was significantly increased by the incubation with various tested inhibitors (Figure 1B) in comparison with the control group without inhibitor treatment. The above findings indicated cytotoxicity and cytostatic effect of Smp24 against A549 cells.

### 2.2. Smp24 Is Internalized into A549 Cells via Endocytosis

Smp24 has been reported to be a membrane lytic peptide [11,12] and the changes in cell surface electrostatics under different conditions reflect cellular phenomena, such as adhesion and interaction with peptides [13]. Hence, we examined the interaction between tumor cells and Smp24 by measuring the membrane potential. After incubation of Smp24 (0, 1.25, 2.5, and 5 μM) with A549 cells, an increase in zeta potential from −12.88 to −9.61 mV was observed (Figure 2A), reflecting that the interaction of cationic Smp24 peptide caused an increase in the net charge of the cell membrane surfaces. Furtherly, we investigated whether Smp24 could be internalized into A549 cells. As presented in Figure 2B, the increasing fluorescence in the cells treated with FITC-labeled Smp24 indicated the internalization of Smp24 into A549 cells in a concentration- and a time-dependent manner, which is consistent with the cell-penetrating property reported in our previous study [10]. Anionic heparan sulfate is an essential component of the cancer cell membrane and the extracellular matrix, which contributes a crucial role in the interaction between the cell membrane and a cell-penetrating peptide [14]. Hence, we further investigated whether the cellular endocytosis of Smp24 was affected by heparan sulfate. As presented in Figure 2C, the decrease in internalization of 5 μM FITC-labeled Smp24 was approximately 1.43%, 46.12%, and 68.82% after 1 h pretreatment with 5, 10, and 20 μg/mL heparan sulfate, respectively, while compared with the control group. Thereafter, we examined the role of energy in the translocation of Smp24. As the common endocytic inhibitors, ammonium chloride alters the pH of acidic endocytic vesicles, while sodium azide can directly eliminate ATP production within the cell membrane [15]. As a result, pretreatment of A549 cells with 40 μM NaN_3_ and 50 mM NH_4_Cl for 1 h obviously inhibited the cellular uptake of Smp24 (Figure 2D,E). As a further confirmation assay, we tested the effect of temperature on the cellular endocytosis (Figure 2F). After 1 h of incubation at 37 °C, the cellular uptake of FITC-labeled Smp24 was approximately 42.89% higher than that at 4 °C, suggesting an energy-dependent and thermo-sensitive endocytosis of Smp24 across A549 cell membrane. Notably, compared to the results of 1 h incubation, more Smp24 was internalized into cells after co-incubation for 6 h, as evidenced by the increasing fluorescence (Appendix A). What’s more, pre-incubation of A549 with heparan sulfate and endocytic inhibitors for 6 h could not decrease the cellular uptake of Smp24 (Appendix A). Furtherly, the cellular uptake of Smp24 at 4 °C and 37 °C after 6 h of incubation was approximately 6.51% and 96.07%, respectively (Appendix A).

### 2.3. Smp24 Promotes ROS Production and Mitochondrial Membrane Potential Decrease

Based on the critical impact of cellular ROS in cell proliferation regulation, we examined whether Smp24 induced ROS production in human lung cancer cells. As presented in Figure 3A, the cell fluorescence intensity was significantly increased in a concentration-dependent manner in Smp24-treated A549 cells, while compared with the control group. In addition, the co-treatment with antioxidant NAC significantly decreased the ROS contents induced by Smp24 (Figure 3A). The obtained results demonstrated that Smp24 elevates the ROS levels in A549 cells.

ROS production is closely associated with mitochondrial membrane potential. Hence, the influence of Smp24 on mitochondrial membrane potential of A549 cells was determined using JC-1 staining. In the control group, red fluorescence was significantly observed, while a remarkable transformation from red to green staining in a concentration-dependent manner was marked in the Smp24-treated cells. What’s more, NAC could attenuate the green fluorescence intensities in the cells stimulated by 5 µM Smp24 (Figure 3B). All these results demonstrated that treatment with Smp24 results in the depolarization of the mitochondrial membrane potential and inducing ROS accumulation in A549 cells.

### 2.4. Smp24 Induces Mitochondrion-Mediated Apoptosis in A549 Cells

ROS production is vital in the apoptosis process in cancer cells [16]. As shown by DAPI staining in Figure 4A, significant reduction in A549 cells size were observed after 24 h of Smp24 treatment, with typical apoptotic features, such as losing nuclear integrity, shrunken nuclei, formation of apoptotic bodies and chromatin condensation, indicating that the growth inhibition of A549 cells induced by Smp24 was associated with apoptosis. Consistently, the proportion of Annexin V-FITC/PI-positive apoptotic cells were remarkably increased by approximately 5.64% to 36.76% when cells were treated with Smp24 (0–5 µM) (Figure 4B, panels a–e). Besides, the presence of NAC significantly attenuated Smp24-induced apoptosis from approximately 36.76% to 19.80% (Figure 4B, panel f).

Due to the important role in apoptosis and necrotic cell death, the mitochondria-mediated apoptotic pathway was investigated to define the mechanism of apoptotic signaling in A549 cells stimulated by Smp24 [16]. As shown in Figure 4C–F, cleaved caspase-3, cleaved caspase-9 and cleaved PARP expression were raised in concentration-dependent manner (Figure 4C–F). Bcl-2 and Bax, which belong to the Bcl-2 family, can regulate the release of mitochondrial proteins, which are closely related to apoptosis. In agreement, an increase in the expression of the pro-apoptotic protein Bax and a decrease in that of the anti-apoptotic protein Bcl-2 were observed in the presence of Smp24 (Figure 4E,F). It is well known that the rise in Bax/Bcl-2 ratio can accelerate cytochrome c release, consequently inducing apoptosis. In agreement, the expression of cytochrome c leaked from mitochondria was dose-dependently increased, following treatment with Smp24 (Figure 4E,F). Together, our data coincidentally demonstrated that Smp24 activates mitochondria apoptotic pathway.

### 2.5. Smp24 Arrests Cycle Distribution of A549 Cells in S Phase and G2/M Phase via Expression Regulation of Phase-related Proteins 

Cell cycle distribution in Smp24-treated A549 cells was investigated to explore whether the proliferation inhibition induced by Smp24 was related to cell cycle arrest. As shown in Figure 5A, when compared with the control group, Smp24 (1.25, 2.5 and 5 µM) increased the number of A549 cells accumulating in S phase from 11.78% to 13.68%, 15.76% and 23.71% and in the G2/M phase from 8.08% to 8.23%, 10.2% and 27.04% after co-incubation for 24 h, respectively (Figure 5A). Consistently, in the G0/G1 phase, a decrease in the ratio of Smp24-treated cells was also observed from 76.00% to 68.2%, 56.5%, and 39.65%. These results indicated that treatment with Smp24 induces the S and G2/M phase arrest in A549 cells.

To identify the underlying mechanism of cell cycle accumulation induced by Smp24, the expression alteration of crucial cell cycle regulation proteins was investigated. In line with its effects on cell cycle arrest, Smp24 upregulated the levels of Cyclin A2, Cyclin B1, p53, and the cyclin-dependent kinase inhibitor, p21^Waf1/Cip1^ but downregulated the levels of Cyclin E1 and CDK2 in concentration-dependent manner, following 24 h treatment. In detail, compared with the control group, 5 µM Smp24 caused approximately a 0.85-, 1.22-, 2.20-, and 2.28-fold rises of Cyclin A2, Cyclin B1, p53, and p21^Waf1/Cip1^ contents in A549 cells, while there were approximately 0.15-, 0.36-, and 0.59-fold and 0.13-, 0.31-, and 0.45-fold declines of CDK2 and Cyclin E1 contents in A549 cells after being exposed to Smp24 (1.25, 2.5 and 5 µM) for 24 h (Figure 5B,C).

### 2.6. Smp24 Induces Autophagy in A549 Cells via Inhibition of the PI3K/Akt/mTOR/FAK and p38/ERK/JNK Signaling Pathways

The formation of autophagosomes in Smp24-treated A549 cells, the hallmark of autophagy, was examined with TEM. As shown in Figure 6A, the control cells displayed normal cytoplasmic organelle and uncondensed chromatin, while Smp24-treated cells contained the increased number of vacuoles and autophagosomes with damaged cellular organelles. To elucidate the underlying molecular mechanisms of Smp24-induced autophagy in A549 cells, the expressions of autophagy-associated protein were measured. In comparison to the control cells, when A549 cells were treated with Smp24 (1.25, 2.5 and 5 µM) for 24 h, the expressions of phosphorylated Akt and mTOR as well as the total Akt were decreased in a concentration-dependent manner, which contrasted sharply with the expression trend of LC3A/B-I/II (Figure 6B,C), reflecting that Smp24 could induce autophagosome formation. Surprisingly, with the increasing concentration of Smp24, the contents of p62, a selective receptor of autophagy substrates, were increased first and then declined (Figure 6B,C).

PI3K and p38 are the upstream of the Akt/mTOR pathway and play vital roles in regulation of cell growth, cycle, apoptosis, migration, and survival [17]. As shown in Figure 6D,E, Smp24 dramatically decreased the phosphorylation of PI3K and p38 in a concentration-dependent manner, while having no effect on total p38 and PI3K expression in A549 cells. The ratios of p-p38/p38 and p-PI3K/PI3K respectively declined to approximately 32.40% and 5.66% following exposure to 5 µM Smp24 (Figure 6D,E).

The phosphorylation of ERK, JNK, and FAK are also important to regulate cancer cell adhesion, invasion, migration and proliferation [18,19]. In comparison to the control cells, Smp24 did not affect the expressions of the total FAK, JNK, and ERK, but it significantly decreased their phosphorylation in a concentration-dependent manner (Figure 6F,G). These findings indicated that Smp24 might induce autophagy in A549 cells via inhibiting the PI3K/Akt/mTOR/FAK and p38/ERK/JNK signaling pathways.

## 3. Discussion

Scorpion-derived peptides have become a rich source for new drug development against various diseases due to their multiple capabilities, especially in cancer treatment [20]. We previously reported the antitumor effect of Smp24, a venom-derived AMP of *Scorpio maurus palmatus*, against A549 human lung cancer cells via damaging the membrane and cytoskeleton. In agreement with the reported study, Smp24 significantly reduces the proliferation of A549 cells. Furthermore, Smp24 is internalized into A549 cells and subsequently interacts with mitochondria, leading to its dysfunction and ROS accumulation.

It is generally accepted that the existence of exclusive anionic components in the cancer cell membranes, such as phosphatidylserine, sialylated gangliosides, O-glycosylated mucins, and heparan sulfate, is associated with the cancer-selective toxicity of cationic AMPs [21]. The electrostatic binding of AMPs to cancer cells are enhanced by the interaction between cationic residues of peptides and anionic components in cell membranes. In line with this, Smp24 significantly increases the zeta potential of A549 cells in a concentration-dependent manner. Furthermore, the presence of heparan sulfate, ammonium chloride, and sodium azide markedly suppresses the cellular uptake of Smp24. These findings suggest the role of endocytosis in the internalization of Smp24 into A549 (Figure 2).

Mitochondria are vital for cancer development and their dysfunction can reduce cancer metabolism [15]. It is well known that the internalization of AMPs into cancer cells are followed by interaction with the mitochondrial membrane and forming of the transition pore, which accordingly induces swelling and rupture of mitochondria due to penetration of cytosolic ions and solutes into the inner membrane. Consequently, a series of events, such as the decline of mitochondrial membrane potential, elimination of ATP generation, accumulation of ROS, and damage of mitochondria happen [16]. Therefore, some AMPs can irreversibly lead to cell death via regulating the mitochondrial pathway [2,16]. For instance, in addition to the membrane pore formation mechanism [22], melittin might induce mitochondrial membrane depolarization, leading to the overproduction of pro-apoptotic factors, such as cytochrome c and ROS, and causing oxidative damage within the cell [5]. Similarly, Smp24 dramatically decreases mitochondria membrane potential (Figure 3B) and accumulation of ROS in a concentration-dependent manner in A549 cells (Figure 3A). The accumulated ROS can induce the expression of p53, which has substantial effects on the initiation of apoptosis via transactivating pro-apoptotic proteins (e.g., Bax) or interacting with anti-apoptotic mitochondrial proteins (e.g., Bcl-2) [23,24]. In agreement, Smp24 upregulates the p53, Bax, and cytochrome c while downregulating Bcl-2 in a concentration-dependent manner (Figure 4E,F and Figure 5B,C), and caspase inhibitors, including z-VAD-FMK and Z-DEVD-FMK, as well as necroptosis inhibitors, such as necrostatin-1, can reduce the suppressive effects of Smp24 on the viability of A549 cells (Figure 1B). Furthermore, after short-term treatment, NAC inhibits the accumulation of ROS in A549 cells (Figure 3A) and reverses Smp24-induced apoptosis (Figure 4B). Thus, Smp24 can induce the apoptosis of A549 cells involved in the mitochondrial pathway and ROS production. Excessive ROS can lead to cell death through necrosis [16], including major characteristics, such as swelling and dysfunction of the cytoplasm and the mitochondrial matrix, chromatin condensation, poration of the cellular membrane and effusion of the cytoplasmic contents into the extracellular space [25]. These findings coincide with non-specific mitochondrial membrane disruption and necrotic cell death in A549 cells caused by Smp24 reported earlier [10].

The crucial role of p53 in regulating the expression of various proteins responsible for both the G1/S and the G2/M transitions was reported earlier [26]. For example, p21^Waf1/Cip1^ might serve as a negative modulator of the G1/S transition by inhibiting the kinase activity of CDK2 complexes [27]. Further, the accumulation of G2/M phase in cell cycle distribution is caused by the binding of p21^Waf1/Cip1^ to the CDK1/cyclin B compound [28]. In addition, CDK2, cyclin A, cyclin B, and cyclin E participate regulation of G1 through early S phase and G2/M phase, respectively [29]. In our experiments, we have observed that the ratios of A549 cells in the S and G2/M phase are enhanced in a dose-dependent manner (Figure 5A). In addition, Smp24 significantly increases the expressions of Cyclin A2, Cyclin B1, p53, and p21^Waf1/Cip1^, while inhibiting the expression of Cyclin E1 and CDK2 (Figure 5B,C), which are consistent with cell cycle arrest phenomenon.

It is well known that there is a complicated association between autophagy and apoptosis, which share common regulation proteins. For instance, PUMA can simultaneously induce apoptosis and autophagy via mitochondrial Bax pathway in return for mitochondrial perturbation [30]. Notably, various AMPs have been reported to exert antitumor activity via concurrently inducing cell apoptosis as well as autophagy processes, such as FK-16 [31], CTLEW [32], and brevenin-2R [33]. In the current study, necrostatin-1, a well-known inhibitor that suppresses both autophagy and apoptosis, significantly decreases the suppressive effects of Smp24 on A549 cells viability. Consistently, the downregulation in the expression of mTOR and Akt is observed in Smp24 treatment, while that of LC3A/B- II/I is significantly enhanced (Figure 6B,C). Interestingly, the protein expression of p62, a substrate of autophagy, is increased by treatment with 1.25 μM Smp24 but decreases in the higher concentration of Smp24 (Figure 6B,C). This phenomenon might be due to the boosting of autophagic flux of Smp24 at a low concentration, while high-concentration Smp24 has more potent effects on degrading p62 than promoting the autophagic flux. The MAPK signaling pathways also contribute a major role in both the autophagy and apoptosis process [34,35], and some compounds have been reported to exert antitumor effects against the A549 cell via inducing apoptosis and autophagy by suppressing both MAPK and PI3K/Akt/mTOR signaling pathways [36]. Consistently, the suppression of MAPK signaling pathways caused by Smp24 is observed in the current study. Thus, Smp24 exerts its cytotoxicity toward human lung cancer cells via inducing apoptosis and autophagy, too.

## 4. Conclusions

In summary, the current study reveals the mode and molecular mechanisms of Smp24 targeting mitochondria in A549 cells. Smp24 is internalized into A549 cells and interacts with mitochondria, leading to the loss of mitochondrial membrane potential, accumulating ROS, causing mitochondrial dysfunctions, which subsequently result in apoptosis, cell cycle arrest, and autophagy. Our findings further expose the antitumor mechanism of Smp24 against NSCLC A549, suggesting its application in lung carcinoma therapy.

## 5. Materials and Methods

### 5.1. Chemicals and Cell Culture

Phosphate-buffered saline (PBS), fetal bovine serum (FBS), RPMI-1640, and trypsin were obtained from Gibco (Grand Island, NY, USA). The A549 cell line was obtained from the American Type Culture Collection (Manassas, VA, USA) and was cultured in RPMI-1640 medium containing 10% FBS and 1% penicillin-streptomycin at 37 °C in an atmosphere of 5% CO_2_. Cytochrome c, Bax, Bcl-2, PARP, cleaved PARP, caspase-3, cleaved caspase-3, caspase-9, cleaved caspase-9, p53, p21, Cyclin A2, Cyclin E1, Cyclin B1, CDK2, Erk, p-Erk, JNK, p-JNK, p38, p-p38, mTOR, p-mTOR, Akt, p-Akt, FAK, p-FAK, PI3K, p-PI3K, p62, LC3A/B- I/II, β-actin, GAPDH and all secondary antibodies were acquired from Cell Signaling Technology (Beverly, MA, USA). Necrostatin-1 (Nec-1, an inhibitor of RIP1K), Z-VAD-FMK (an inhibitor of pan caspase), Z-DEVD-FMK (an inhibitor of caspase-3), N-Acetyl-L-cysteine (NAC), mitochondrial membrane potential assay kit for JC-1, reactive oxygen species (ROS) assay kit, cell cycle and apoptosis analysis kit, and DAPI were obtained from Beyotime Institute of Biotechnology (Shanghai, China). Smp24 and FITC-labeled Smp24 were synthesized as reported in our previous study [11].

### 5.2. Cell Viability and Proliferation Assays

Cellular viability was analyzed via MTT method as reported in our previous study [37]. Briefly, A549 cells (1 × 10^4^ cells/well) were cultured in 96-well plates and treated with a gradient concentration of Smp24 (1.25–20 μM) at different time intervals (12, 24, and 48 h). A549 cells were pre-incubated with different inhibitors including 40 μΜ Nec-1, 40 μM Z-DEVD-FMK, 40 μM Z-VAD-FMK, and 2 mM NAC for 30 min before 5 μM Smp24 was incubated with the cells for 12 h to investigate the responsibility of different signaling pathways in the cytotoxic effects of Smp24. After incubation, 10 µL MTT (5 mg/mL) was then added and incubated for 4 h at 37 °C in the dark. Subsequently, 200 µL DMSO was applied into each well after discarding the cell medium, and the cell viability was estimated by calculating the absorbance value at 490 nm via a microplate reader (Tecan Company, Männedorf, Switzerland). The experiments were performed in triplicate.

### 5.3. Peptide Internalization Analysis

After being seeded onto a 24-well plate overnight (1 × 10^5^ cells/well), A549 cells were subsequently incubated with 1.25, 2.5, and 5 μM of FITC-labeled Smp24 at 37 °C for 1 h and 6 h. Flow cytometry (Becton Dickinson Company, Bedford, MA, USA) was used to measure the cell fluorescence intensity after washing with PBS. The location of Smp24 within cells was observed with fluorescence microscope at 400 × magnification. In detail, the cells were washed with PBS after treatment with 5 μM FITC-labeled Smp24 at 37 °C for 6, 12, and 24 h, respectively. Cells were then fixed with 4% paraformaldehyde (PFA) for 30 min, followed by staining with DAPI for 10 min. Approximately three single-plane pictures of each well were obtained.

To identify the effects of heparan sulfate on peptide internalization, 5 µM FITC-labeled Smp24 was pre-incubated with 5, 10, or 20 µg/mL heparan sulfate in RPMI-1640 medium for 30 min before the mixture was incubated with cells (1 × 10^5^ cells/well). After 1 h and 6 h, flow cytometry (Becton Dickinson Company, Bedford, MA, USA) was used to measure the cell fluorescence intensity.

To identify whether the cellular energy state affects internalization, A549 cells were placed at 4 °C or 37 °C for 30 min before incubated with 5 µM FITC-labeled Smp24 for 1 h and 6 h. Thereafter, the cells were harvested, washed, and re-suspended in 200 µL of PBS for flow cytometry analysis. As a further confirmation of the role of energy in the translocation of Smp24, the FITC-labeled Smp24-treated A549 cells were pre-incubated with 40 μM NaN_3_ or 50 mM NH_4_Cl for 30 min and then subjected to flow cytometry. All experiments were detected in triplicate.

### 5.4. Zeta Potential Measurement

Zeta potential analysis was performed to evaluate the interaction between Smp24 and cancer cells. In short, after re-suspended in PBS, 1 × 10^5^ A549 cells were mixed with Smp24 (0, 1.25, 2.5, and 5 µM). Then, the mixtures were loaded into Folded Capillary cell (DTS1070, Malvern Instruments Ltd., Worcestershire, UK) for measurement of zeta (ξ) potential with the Zetasizer system (Nano ZS; Malvern Instruments Ltd., Worcestershire, UK). The A549 cell suspension without Smp24 was considered as a control. All experiments had been detected in triplicate.

### 5.5. Transmission Electron Microscopy Analysis

After cultivation in a 6-well plate (2 × 10^5^ cells/well) for 24 h, A549 cells were co-incubated with 5 μM Smp24 for another 24 h. Negative controls were defined as cells without Smp24 treatment. The cells were harvested and fixed with electron microscope fixative (2.5% glutaraldehyde at 0.1 M PBS) at room temperature for 2 h and then at 4 °C for 48 h. After removing the fixative, cells were incubated with 8% sucrose in PBS, followed by post-fixation with 1% osmium tetraoxide for 1 h at 4 °C. The cells were subsequently washed with PBS three times for 10 min. After being dehydrated with a series of gradient ethanol/water solutions, the cells were embedded in Poly/Bed 812 resin (Pelco, Redding, CA, USA). Lead citrate was used to stain the ultrathin sections, and samples were examined on a transmission electron microscope (TEM, Hitachi Company, Tokyo, Japan) at 80 kV.

### 5.6. Fluorescence Microscopy Analysis

For intracellular ROS content measurement, after being pretreated with Smp24 (0, 1.25, 2.5, and 5 μM) for 12 h, A549 cells were then washed with PBS before incubation with 10 μM of 2, 7-dichlorodihydro-fluoresceindiacetate (DCFH-DA) at 37 °C for 30 min in the dark. Inverted fluorescence microscopy (Axio Observer, Zeiss, Oberkochen, Germany) was used to observe cells at 200 × magnification. NAC was used as the positive control.

The changes on the mitochondrial membrane potential of A549 cells caused by Smp24 were identified using JC-1 staining (Beyotime Institute of Biotechnology, Shanghai, China). After being incubated with different concentrations of Smp24 for 12 h, A549 cells were then stained with JC-1 for 30 min at 37 °C according to the manufacturer’s instructions. The changes were observed under fluorescence microscopy (Axio Observer, Zeiss, Oberkochen, Germany) at 400 × magnification. NAC was use as the positive control and approximately three random photos of each well were captured.

### 5.7. Cell Cycle and Apoptosis Analysis

Flow cytometry was used to identify the effects of Smp24 on cell cycle distributions and apoptosis in A549 cells. A549 cells with 2 × 10^5^/well in density were cultured in 6-well plates overnight and treated with a gradient concentration of Smp24 or 2 mM NAC for 24 h. After being collected by centrifugation, cells were subsequently washed with cold PBS and followed by fixing with 70% ethanol overnight at 4 °C. The cells were stained with PI and RNase A at 37 °C for 30 min after being washed with cold PBS. For the apoptosis analysis, harvested cells were stained with Annexin V-FITC and PI for 15 min at room temperature and then detected by flow cytometry (Becton Dickinson Company, Bedford, MA, USA). All experiments were conducted in triplicate.

### 5.8. Western Blot Analysis

After culturing in 6-well plates, the 2 × 10^5^ A549 cells were incubated with Smp24 (0, 1.25, 2.5, and 5 μM) for 24 h. After treatment, the harvested cells were extracted with RIPA lysis solution containing 1% phosphatase and protease inhibitors (FDbio Science Biotech Co., Ltd., Hangzhou, China) on ice for 15 min and then the supernatant was used for SDS-PAGE analysis and transferred to polyvinylidene difluoride (PVDF) membrane (Millipore, Billerica, MA, USA). After mixing with appropriate primary antibodies (1:1000) at 4 °C overnight and horseradish peroxidase-conjugated secondary antibodies (1: 2000) at 25 °C for 1 h, blot bands were observed by the hypersensitive ECL chemiluminescence agent and calculated using Image J software (64-bit, National Institutes of Health, MD, USA) in triplicate.

### 5.9. Statistical Analysis

All data were presented as mean ± SEM. Data were analyzed using one-way ANOVA followed by Bonferroni’s test through GraphPad Prism 5.0 (GraphPad Software, Inc., La Jolla, CA, USA). Statistical significances were shown as * *p* < 0.05, ** *p* < 0.01, and *** *p* < 0.001.

## Figures and Tables

**Figure 1 toxins-14-00590-f001:**
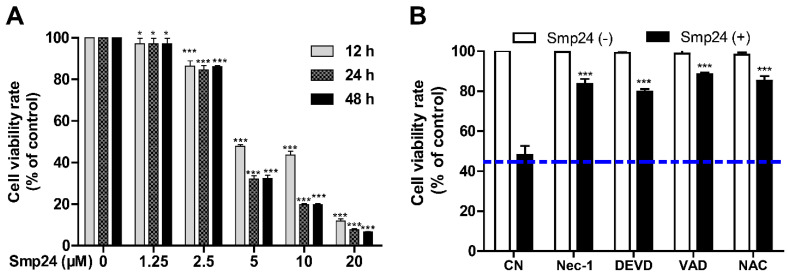
Effect of Smp24 on proliferation of A549 cells. (**A**) Viability of A549 cells treated with different concentrations of Smp24 for 12, 24, and 48 h. Negative control: cells treated with equivalent solvent for corresponding time. (**B**) Effects of inhibitors on the viability of Smp24-treated A549 cells. A549 cells were treated with 5 μM Smp24 for 12 h after being pre-incubated with the inhibitors 40 μM necrostatin-1 (Nec-1), 40 μM Z-DEVD-FMK (DEVD), 40 μM Z-VAD-FMK (VAD), and 2 mM NAC for 30 min. Negative control: cells were treated with above condition without Smp24. Bule dashed line represents the mean value of cell viability rate after treatment with Smp24 for 24 h. Data are normalized to control and presented as mean ± SEM (*n* = 3). * *p* < 0.05 and *** *p* < 0.001 are considered statistically significant when compared with the control group.

**Figure 2 toxins-14-00590-f002:**
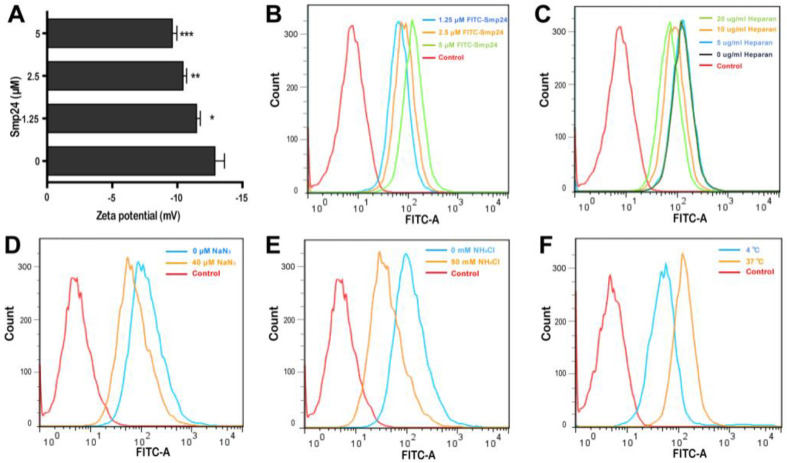
Internalization of Smp24 into A549 cells. (**A**) Effect of Smp24 on the zeta potential of A549 cells. Negative control: cells treated with equivalent solvent. **(B**) The internalization of FITC-labeled Smp24 into A549 cells after 1 h of treatment. (**C**–**F**) Effect of heparan sulfate, NaN_3_, NH_4_Cl, and temperature on the intracellular uptake of FITC-labeled Smp24 for 1 h, respectively. Negative control: cells treated with equivalent free-FITC solution. The values are presented as mean ± SEM (*n* = 3). * *p* < 0.05, ** *p* < 0.01, and *** *p* < 0.001 are considered statistically significant compared to the control group.

**Figure 3 toxins-14-00590-f003:**
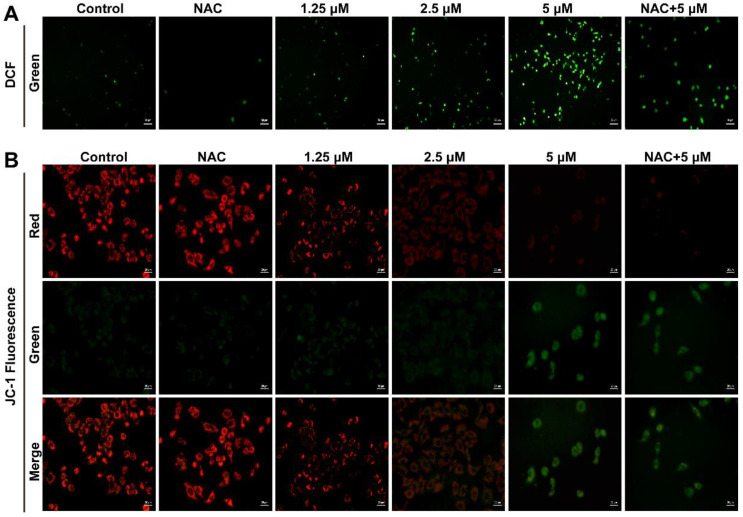
Influence of Smp24 on ROS production and mitochondrial membrane potential of A549 cells. (**A**) Characteristic fluorescence photographs of A549 cells stained with DCFH-DA. Scale bar, 50 μm. (**B**) Representative JC-1 fluorescence photographs of A549 cells. Scale bar, 20 μm. Negative control: cells treated with equivalent solvent for corresponding time.

**Figure 4 toxins-14-00590-f004:**
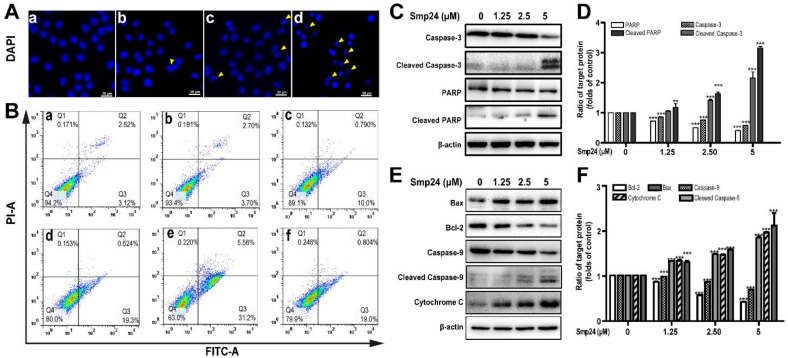
Apoptosis of A549 cells induced by Smp24. (**A**) Morphology of apoptotic A549 cells treated with Smp24 for 24 h. The nucleus was stained with DAPI and observed under fluorescence microscopy. Panel a: the control cells; Panels b–d: A549 cells in the presence of Smp24 (1.25, 2.5, or 5 µM), respectively. The apoptotic bodies are indicated by yellow triangles. Scale bar, 20 μm. (**B**) Representative cytometry analysis of apoptotic A549 cells after treatment with Smp24 or NAC for 24 h. Panel a: the control cells; Panels b–f: A549 cells in the presence of NAC, Smp24 (1.25, 2.5, and 5 µM), or NAC + 5 µM Smp24, respectively. (**C**, **E**) Representative western blots of caspase-3, cleaved caspase-3, PARP, cleaved PARP, Bax, Bcl-2, caspase-9, cleaved caspase-9 and cytochrome c. (**D**, **F**) Quantification of band densities in C, E. Bars represent the ratio of the target protein to the control group. Negative control: cells treated with equivalent solvent. Band densities were analyzed by Image J software and values were presented as mean ± SEM (*n* = 3). ** *p* < 0.01 and *** *p* < 0.001 are considered statistically significant when compared with the control group without Smp24.

**Figure 5 toxins-14-00590-f005:**
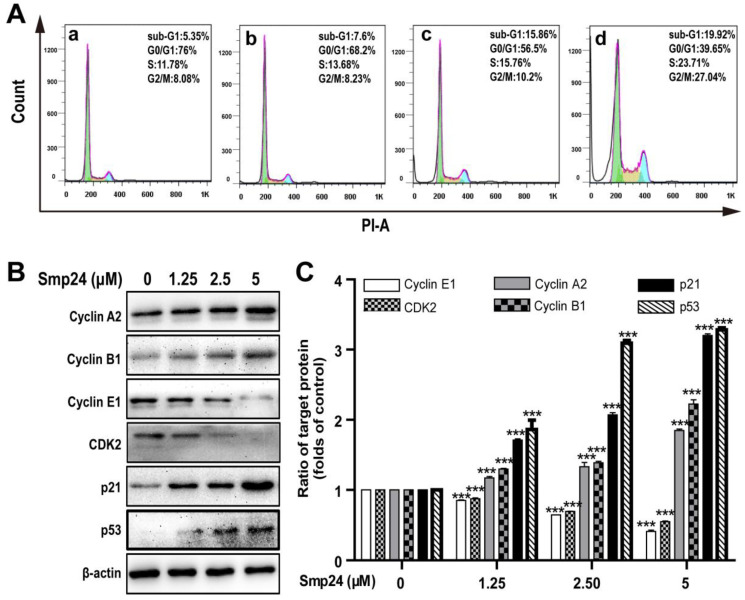
Effect of Smp24 on cell cycle distribution in A549 cells. (**A**) Flow cytometry analysis of cell cycle stages. A549 cells were exposed to Smp24 (1.25, 2.5, and 5 µM) for 24 h and followed by analysis with flow cytometry. Panel a: the control cells; Panels b–d: A549 cells in the presence of Smp24 (1.25, 2.5, or 5 µM), respectively. (**B**) Representative western blots of Cyclin A2, Cyclin B1, Cyclin E1, CDK2, p21^Waf1/Cip1^, and p53. (**C**) Quantification of band densities in B. Bars represent the ratio of the target protein to the control group. Negative control: cells treated with equivalent solvent for corresponding time. Data are presented as mean ± SEM (*n* = 3). *** *p* < 0.001 are considered statistically significant while compared with the control group without Smp24.

**Figure 6 toxins-14-00590-f006:**
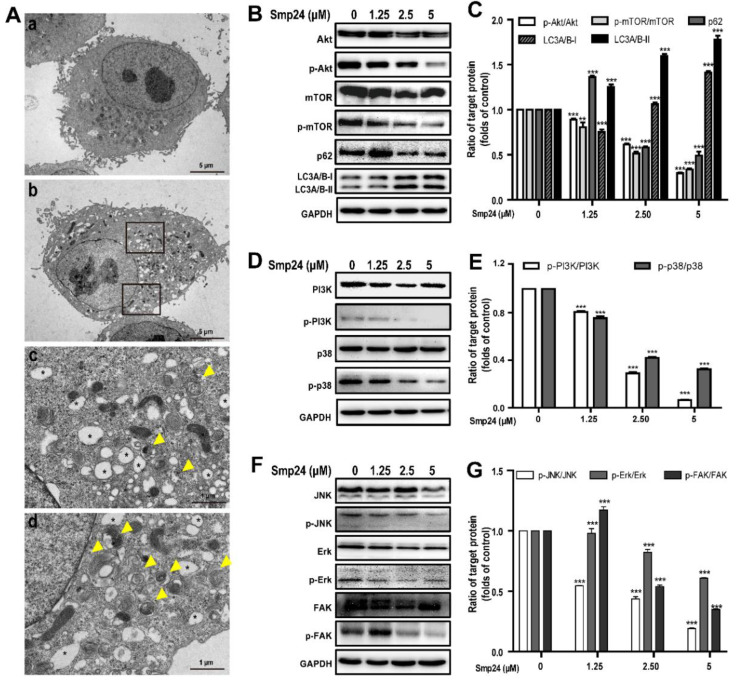
Regulation of autophagy and signaling pathways by Smp24. (**A**) TEM analysis of morphological structure of A549 cells with Smp24 treatment for 24 h. Panel a: the control cells, Panel b: A549 cells in the presence of 5 µM Smp24. Panels c and d: the magnified local areas in the corresponding upper and lower black squares of panel b. The vacuoles and autophagosomes were marked by black asterisk and yellow triangles, respectively. Magnification: 1200 × in panels a and b, 6000 × in panels c and d. (**B**,**D**,**F**) Representative western blots of proteins belonging to PI3K/Akt/mTOR/FAK and p38/ERK/JNK signaling pathways. (**C**,**E**,**G**) Quantification of band densities in B, D, and F. Bars represent the ratio of the target protein to the control group. Negative control: cells treated with equivalent solvent for corresponding time. Data are presented as mean ± SEM (*n* = 3). ** *p* < 0.01 and *** *p* < 0.001 are considered statistically significant as compared to the control group without Smp24.

## Data Availability

All data are available on request.

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
