# Peer review of "The Strong Anti-Tumor Effect of Smp24 in Lung Adenocarcinoma A549 Cells Depends on Its Induction of Mitochondrial Dysfunctions and ROS Accumulation"

_toxins, 2022, doi:10.3390/toxins14090590_

Round 1
Reviewer 1 Report
The manuscript entitled: “The strong anti-tumor effect of Smp24 in lung adenocarcinoma A549 Cells depends on its inductions of mitochondrial dysfunctions and ROS accumulation” is a thorough work done for the testing of the antimicrobial peptide (AMP) Smp24 from scorpion venom. The study was carried out with a variety of experiments that shows the anti cancer activity and efficiency as well as the mechanism of action of this AMP against A549 lung cancer cells. However, there are some points that may need some clarification:
- For typos, writing errors and so on, I have only detected one possible error in line 86: it says 40 Mm Nec-1. Does it mean 40 mM? or perhaps 40 mM instead?
- In the introduction, (lines 25 to 27) the authors mention: “the cytotoxicity and drug resistance caused by those traditional therapy leads to the urgent need of alternative treatment with low cytotoxicity and treatment resistance”. It is known that AMPs have specific mechanisms of action that minimize drug resistance and the authors mention that in the abstract, the introduction (line 32) and also they talk of mitochondrial membrane disruption (lines 53, 62), etc. However, cytotoxicity is more difficult to control. It is also well known that there are differences in the composition of cell membranes of bacteria with eukaryotic cell membranes or also in the case of cancer cells with non-cancer cells which the authors also mention in the conclusions (lines 351 to 252) and references 22 and 34. However, one thing that it is missing in this study is a probe of toxicity of their AMP, that is, the testing of their Sm24 peptide against normal cells.
The authors report controls for each experiment of this work but these controls are all done with A549 cancer cells but none with healthy cells. The authors do not need to show every single experiment done for healthy lung cells but at least, I may suggest, they can present one experiment (perhaps 3.1) that clearly shows potential or negligible toxicity of their AMP Smp24 against healthy lung cells.
- There is a lot of work presented in this manuscript for only one single peptide and the results are certainly significant for those interested in the study of mechanisms of action of AMPs. Just for that, it is fine for publication. However, if it turns out that Smp24 is toxic for healthy lung cells, the authors conclusion that their peptide has application in lung carcinoma therapy will not be possible and the authors would have to change that statement and focus their work for the mechanistic results, which again, they are significant by themselves.
Reviewer 2 Report
The manuscript, which title is The strong anti-tumor effect of Smp24 in lung adenocarcinoma A549 Cells depends on its inductions of mitochondrial dys-functions and ROS accumulation, is interesting and novelty. However, there are several questions in the manuscript. Please indicate the brand and origin of the machines. The authors should provide the negative or relative inhibited control group in the manuscript. The authors should provide mechanism of the peptide Smp24 into the cells. The authors should provide the evidence about the concentration and treatment duration of Smp24 in the manuscript.
Round 2
Reviewer 1 Report
Since the suggested information and pointed errors have been corrected by the authors, the manuscript entitled: “The strong anti-tumor effect of Smp24 in lung adenocarcinoma A549 Cells depends on its inductions of mitochondrial dysfunctions and ROS accumulation”, can be considered for publication.
Reviewer 2 Report
No more question